# Introgression and mating patterns between white-handed gibbons (*Hylobates lar*) and pileated gibbons (*Hylobates pileatus*) in a natural hybrid zone

**Kazunari Matsudaira**[1¤]*, **Ulrich H. Reichard**[2], **Takafumi Ishida**[3¤], **Suchinda Malaivijitnond**[1,4]

1 Department of Biology, Faculty of Science, Chulalongkorn University, Bangkok, Thailand, 2 Department of Anthropology and Center for Ecology, Southern Illinois University Carbondale, Carbondale, Illinois, United States of America, 3 Department of Biological Sciences, School of Science, The University of Tokyo, Tokyo, Japan, 4 National Primate Research Center of Thailand, Chulalongkorn University, Saraburi, Thailand

¤ Current address: Graduate School of Asian and African Area Studies, Kyoto University, Kyoto, Japan
* kaz.matsudaira@gmail.com

**Data Availability Statement:** All DNA sequence files are available from the DDBJ/ENBL/GenBank

## Abstract

Gibbons (Family Hylobatidae) are a suitable model for exploring hybridization in pair-living primates as several species form hybrid zones. In Khao Yai National Park, Thailand, white-handed gibbons (*Hylobates lar*) and pileated gibbons (*Hylobates pileatus*) are distributed parapatrically and hybridize in a narrow zone. Their phenotypic characteristics suggest limited inter-species gene flow, although this has never been assessed. To uncover the history and degree of gene flow between the two species, we studied the genetic structure of gibbons in the hybrid zone by analyzing fecal DNA samples, phenotypic characteristics, vocalizations and individuals' social status. We determined eight autosomal single nucleotide variant (SNV) loci, and mitochondrial DNA (mtDNA) and Y-chromosomal haplotypes of 72 gibbons. We compared these markers with reference types of wild pureblood white-handed gibbons (*n* = 12) in Kaeng Krachan National Park and pureblood pileated gibbons (*n* = 4) in Khao Soi Dao Wildlife Sanctuary. Autosomal genotypic analyses confirmed the various levels of mixed ancestry for several adult gibbons with or without atypical phenotypic traits in Khao Yai National Park. In some other adult gibbons, the mixed ancestry was not detected in either autosomal SNVs or their phenotypic traits but the mtDNA. Both male and female adult hybrids formed reproductive units mainly with a phenotypic pureblood partner and many of them produced offspring. Taken together, our results suggest that once hybridization occurs, white-handed-pileated-gibbon hybrids can reproduce with either parental species and that the backcrossing and thus introgression may occur in successive generations, with no drastic changes in phenotypic appearance.

database (accession numbers LC633853-
LC633876.)

**Funding:** KM, Grant No. H25-68, JSPS
Postdoctoral Fellowship for Research Abroad,
https://www.jsps.go.jp/english/index.html KM,
Postdoctoral Fellowship under Rachadapisaek
Sompote Fund, The Graduate School,
Chulalongkorn University, https://www.grad.chula.
ac.th/en/ The founders had no role in study design,
data collection and analysis, decision to publish, or
preparation of the manuscript.

**Competing interests:** The authors have declared
that no competing interests exist.

## Introduction

Natural hybridization and introgression are widespread phenomena in animals [1], including primates [2, 3]. Genome-scale studies have revealed "hidden" ancient (or historical) hybridization and introgression among many primate taxa (e.g. chimpanzees and bonobos [4], orangutans [5], baboons [6] and macaques [7]). While genome-scale studies are suitable for uncovering the historical impact of hybridization and introgression in primate evolution, field observations in hybrid zones are essential to investigate proximate mechanisms that prevent (i.e., produce reproductive barriers) or promote hybridization [8]. Integrating field- with lab-based studies can help to illuminate entire hybridization/introgression histories [9]. For example, baboons are one major model of studying hybridization in primates because a long history of field observations [10], genomic studies [6] and an integration of both exists [11, 12].

Gibbons (family Hylobatidae) provide an equally interesting model for studying hybridization. First, among extant apes, only gibbons show ongoing, active hybrid zones [13–15]. Second, their typical reproductive system is monogamous pairs, and they reproduce primarily with the pair-living partner [16, 17], although some variations in social organizations [18–20], mating patterns [19, 21–23] and reproductive patterns [16, 24] have also been observed. Wild gibbons start reproduction at about 10 years of age, and an adult female reproduces only once in about every three years [25]. Considering the low ratio of extra-pair paternity [16, 17, 24], both female and male gibbons reproduce a limited number of offspring into which they invest substantially. The monogamous reproductive pattern seems more sensitive to potential disadvantages of hybridization [26], such as, for example, testicular dysfunction and sterility in hybrid males [27], compared to polygamous and multiparous reproductive system. Finally, gibbon species share basic biological characteristics, such as being adapted to a mainly frugivorous diet and life in the canopy [28]. Competitive inter-species relationships have been suggested for most species [29], which is indirectly supported by the notion of an allopatric distribution of species, except for the sympatric distribution between large-bodied siamangs (*Symphalangus syndactylus*) and smaller gibbons of the genus *Hylobates* [30]. Species differences in gibbons appear in their vocal repertoire and pelage coloration [15], which are important in social communication and probably as well in species identification.

Hybridization between white-handed gibbons (*Hylobates lar*) and pileated gibbons (*Hylobates pileatus*) in Khao Yai National Park, Thailand, has been studied for more than 40 years [13, 15, 31–33]. Observations have uncovered a limited number of hybrid gibbons that have been identified by mixed-species vocalizations and external morphology, such as pelage coloration, distributed in a narrow zone along the Lam Ta Khong River [13]. Early studies have argued that the hybrid population could be a "sink population" where introgression (or gene flow) between the two species is limited [13, 33]. However, since sophisticated molecular methods were unavailable in the 1970s and 1980s, the argument of an existing "sink hybrid population" could not be proven at the molecular level. Recently, we analyzed mtDNA of a population that was thought of as "pure" white-handed gibbons, at the Mo Singto study site in Khao Yai, only to find confirmation that some phenotypically white-handed gibbons possessed mtDNA of a typical pileated gibbon haplotype [34]. This characteristic suggested the presence of several backcross generations and some introgression between the two species even within the Mo Singto subpopulation of phenotypic white-handed gibbons. Moreover, another recent study analyzing genome-wide single nucleotide variants (SNVs) among captive gibbons detected an introgression signal between every combination of the two species individuals (among 20 white-handed gibbons and 10 pileated gibbons), indicative of a relatively long hybridization history [35].

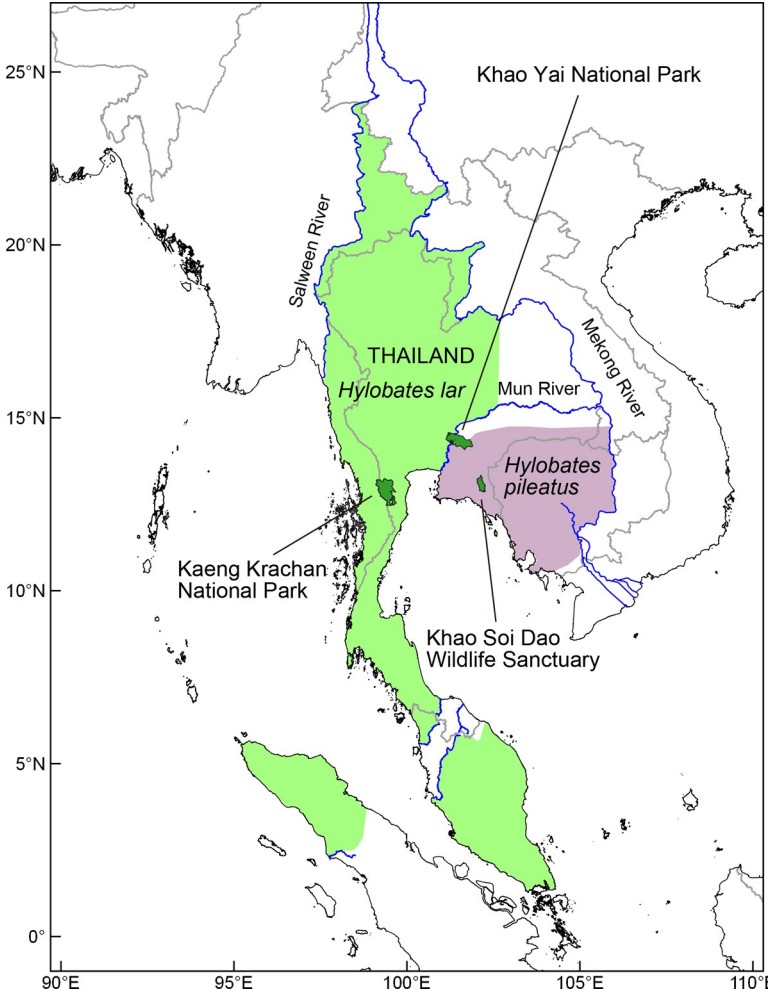

**Fig 1. Distribution of white-handed gibbons (*Hylobates lar*), pileated gibbons (*H. pileatus*) and study sites.** The approximate species distribution range was manually drawn based on previous studies [15, 36] and the Gibbon Research Lab (http://www.gibbons.de/main2/). The protected areas, country borders, coastlines and rivers came from OpenStreetMap and OpenStreetMap Foundation, which was available under the Open Database License.

To fully understand the pattern of hybridization in a long-established hybrid zone, the present study aimed to uncover the degree of introgression and mating patterns between white-handed and pileated gibbons in the ongoing, active natural hybrid zone in Khao Yai National Park, Thailand, one of the oldest National Parks in Thailand which since it's establishment in 1962 has had no meaningful connectivity to other forests in Thailand. We analyzed autosomal SNVs, together with mtDNA and Y-chromosome polymorphisms among gibbons living in the natural hybrid zone and assessed the presence of introgression by analyzing the admixture level of individuals and morphological characteristics. We also compared hybrids with pure-blood white-handed gibbons in Kaeng Krachan National Park (approximately 250 km Southwest of the Khao Yai hybrid zone) and pureblood pileated gibbons in Khao Soi Dao Wildlife Sanctuary (approximately 150 km Southeast of the Khao Yai hybrid zone; Fig 1). Based on the genetic analyses and observations, we assessed hybrid gibbons' mating patterns to uncover their reproductivity and direction of introgression, which is potentially biased between the two species.

## Methods

### Ethical statement

We conducted the study under the permission of the National Research Council of Thailand (NRCT); the Department of National Parks, Wildlife and Plant Conservation of Thailand (DNP); Khao Yai National Park; Kaeng Krachan National Park; and Khao Soi Dao Wildlife Sanctuary (NRCT project No. 2757).

### Study sites and distribution survey

We conducted a field survey in the natural hybrid zone of white-handed gibbons and pileated gibbons in Khao Yai National Park, Thailand (N 14.43872˚, E 101.37238˚) (Fig 1), from February to December 2014. In this study, we selected three survey areas: (1) Mo Singto, (2) the western side of the valley of Lam Ta Khong River and (3) the northern slope of Khao Khiao (Fig 2), where the high potential of the presence of hybrid gibbons was expected based on previous studies and accessibility.

We estimated the presence of gibbon groups and their putative species identity by their prominent vocalization [37]. We set up 21 listening posts each about 500 m apart from one

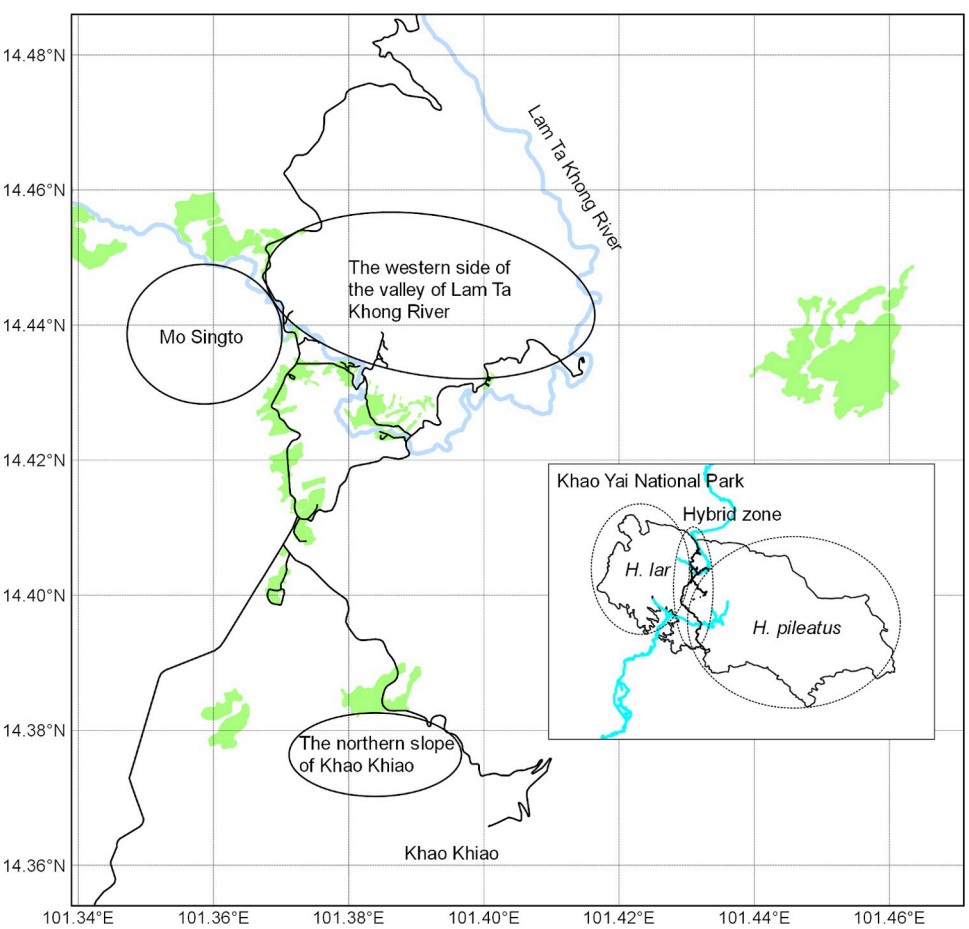

**Fig 2. Three study areas in Khao Yai National Park.** Black solid lines in the main map represent motorways. Light blue lines represent rivers. Light green areas represent grasslands. The border of Khao Yai National Park, roads, rivers and grasslands came from OpenStreetMap and OpenStreetMap Foundation, which was available under the Open Database License.

another along nature trails and motorways. We conducted initial listening survey from only one listening post at a time, spending one to three days at each listening post based on listening conditions. When we noticed gibbons' vocalizations, we recorded start and end times, song category (i.e. duets, OA duets, disturbance-induced songs, male solos and others), the songs' putative species identity, and approximate distance and direction from the listening post to the singing location. In addition to listening post surveys, we also recorded the same gibbon vocalization parameters whenever we were in the forest.

After the initial listening survey, we also directly followed and observed gibbon groups, and collected fecal samples. Whenever we encountered gibbons, we recorded the start and end times of direct observation. We obtained encounter locations from a GPS data logger (GPSMAP60CSx, Garmin, KS) brought by the observer. GPS coordinates were extracted in 5 min intervals and plotted on a map to estimate gibbon groups' home ranges.

We did not know for how long hybridization had occurred in Khao Yai. Thus, we suspected that many white-handed and pileated gibbons' morphospecies in Khao Yai might have some amount of mixed genetic background. Therefore, Khao Yai gibbons could not be considered good comparative sample to represent pureblood white-handed or pureblood pileated gibbons. To avoid this problem in our genetic analysis of identifying hybrids, we selected our comparative samples from populations more than a hundred kilometers distant to the hybrid zone, from deep within each species' distribution range, and conducted field surveys for white-handed gibbons in Kaeng Krachan National Park (12.88499° N, 99.63266° E) from March to April 2015, and for pileated gibbons in Khao Soi Dao Wildlife Sanctuary (13.10296° N, 102.19195° E) from February to April 2016 (Fig 1). These study sites were about 180 km and 130 km linear distance apart, respectively, from each species' closest putative historical species contact zone [38]. Based on the distance far from the hybrid zone and the morphospecies, effects of recent hybridization between the two species were considered negligible.

## Group composition and morphological data collection

When we encountered a gibbon group, we identified each group member's sex and all members' age category: adult (>8 years old and the main adult member), subadult (6–8 years old, >8 years old and not the main adult member), adolescent (4–6 years old), juvenile (2–4 years old) and infant (0–2 years old) (modified after [39]). We recorded the gibbons' morphological characteristics using a digital camera (Lumix FZ-200, Panasonic, Japan).

## DNA sample collection, extraction and quantification

Fecal samples were collected as a source of DNA. We used the well established ethanol–silica two-step method [40]. We kept samples at room temperature at the study sites for up to one year and later stored them at 4°C in a lab refrigerator until they were processed. We also collected some fecal samples in lysis solution by using cotton buds [41]. We collected 129 fecal samples in Khao Yai, 71 fecal samples in Kaeng Krachan, and nine fecal samples in Khao Soi Dao. Because we expected that not every fecal sample would contain sufficient amounts of DNA for genotyping, multiple samples were collected from the same individuals to ensure success in genotyping. Fecal sample collection from non-habituated gibbons was difficult and the number of samples collected for each individual remained limited. Furthermore, we collected hundreds of fecal samples in Khao Yain National Park in previous studies [16, 34]. Thus, we also selected and genotyped DNA samples of adult gibbons from the previous samples. Overall, 88 individuals were genotyped. Of those, 65 adult plus seven non-adult gibbons were genotyped from Khao Yai, 12 adult gibbons were genotyped from Kaeng Krachan and four adult gibbons were genotyped from Khao Soi Dao.

We extracted DNA using a QIAamp Fast DNA Stool Mini Kit (Qiagen, Germany) with some modifications. About 200 mg feces (or 200 mL lysis buffer) were vortex-mixed with 1 mL InhibitEX® Buffer and incubated overnight at room temperature, shaking with a tube mixer. We extended the incubation time with proteinase K for 1 h. Furthermore, the final incubation step with 200 μL elution buffer (Buffer ATE) lasted 30 min.

DNA extracted from fecal samples included host and non-host DNA, such as gastrointestinal microbiota. Since the amount of host DNA was critical for the DNA analysis [42], we thus measured the concentration of host DNA using quantitative real-time PCR targeting *c-myc* [42, 43] for quality control. The reaction was conducted in the same way as in a previous study [16].

## SNV genotyping

We searched for SNVs using two approaches. First, the DNA sequences of autosomal noncoding loci [44] were obtained from GenBank. The data set consisted of 11 white-handed and four pileated gibbons. We selected SNV sites fixed between the two species, and no other SNVs were detected at least 45 bases upstream and downstream of the target SNV sites, where primers and probes were designed. In this process, 14 loci were available from the database and seven of those loci fulfilled the above criteria. Of those, two loci were estimated to be located on the close position at a chromosome, and thus one of them was excluded and six loci were tested for genotyping. Genotyping succeeded in five of the six loci. Second, we also screened autosomal SNV loci by using primers reported by Perelman et al. [45], who determined a sequence of one white-handed gibbon, one agile gibbon (*Hylobates agilis*), one Müller's gibbon (*Hylobates muelleri*) and one siamang. The sequence of 39 autosomal loci was available for the white-handed gibbon sample. Of those, 17 loci held SNVs which were specific to the white-handed gibbon sample and different from the other three gibbon species samples and the genome sequence of a northern white-cheeked gibbon (*Nomascus leucogenys*), nomLeu3/ Nleu_3.0 [46]. In 12 of the 17 loci, SNVs were located in non-coding regions. We determined the sequences of five of the 12 loci in white-handed and pileated gibbon DNA samples from our DNA repository by Sanger sequencing and three loci held suitable SNV sites for designing primers and probes. In both the first and the second approaches, to avoid genotyping linked loci, we checked the approximate location of each SNV in the white-handed gibbon genome by comparing it with the northern white-cheeked gibbon genome sequence [46] and the result of chromosome painting [47]. Based on the sequences, TaqMan probes and primers for SNV loci were designed by Applied Biosystems (MA, United States) (S1 Table).

We genotyped eight SNVs by real-time PCR using TaqMan MGB Probes. Each 10 μL reaction mix contained 1× TaqMan GTXpress Master Mix (Applied Biosystems), 1× TaqMan SNP Genotyping Assay (Applied Biosystems) and 2 μL of DNA template. PCR cycle conditions were as follows: preincubation at 25˚C for 30 s and at 95˚C for 20 s, followed by 45 cycles at 95˚C for 15 s and 60˚C for 1 min. We conducted the reactions with a StepOnePlus Real-Time PCR System (Applied Biosystems) or a QuantStudio 6 Flex Real-Time PCR System (Applied Biosystems). We initially duplicated genotyping of a subset of DNA samples for three SNV loci and obtained the same genotypes for each repetition, except when genotyping failed. This indicates that allelic dropout in our real-time PCR condition was negligible, and thus we conducted only one reaction per locus per sample for further genotyping.

## Ancestry analysis

We estimated each individual's genetic admixture levels by STRUCTURE Version 2.3.4 [48]. We used the admixture model of Pritchard et al. [48] with the number of populations (*K*),

from 2 to 5. We performed 10 independent runs for each *K*. Each run consisted of 1,000,000 generations of Markov chain Monte Carlo with a burn-in period of 100,000 generations. Proportions of each ancestry for each animal were averaged over the 10 independent runs.

## Mitochondrial DNA (mtDNA) and Y-chromosome genotyping

We determined the haplotypes of hypervariable region 1 (HVR1) of the mtDNA (631- or 632-base fragment) and the Y-chromosome haplogroups based on an *sry* indel (2-base indel), as done in previous studies [34, 49]. Based on the 2-base indel in the non-coding region of the *sry* gene, it was possible to distinguish two substantially different sequence groups, which we named haplogroup YA (2 bases longer) and YB (2 bases shorter) [49]. In the study [49], only YA was observed among captive white-handed gibbons, but both haplogroups YA and YB were observed among captive pileated gibbons. The same pattern was reported in another independent study that focused on different regions of the Y-chromosome [50]. We outsourced the sequencing and fragment analyses of PCR products to Macrogen (Korea). The electropherograms of HVR1 were visualized by MEGA6 [51] and each sequence was determined. We made a median-joining network [52] of HVR1 by using PopART 1.7 [53], and assigned haplotypes based on the network drawn. We determined the *sry* PCR product's fragment size using GeneMapper (Applied Biosystems) and assigned haplogroups based on a previous study [49].

## Results

### Mixed ancestry of autosomes in Khao Yai gibbons

The STRUCTURE analysis of the eight autosomal SNVs successfully distinguished white-handed and pileated gibbon ancestry from one another when *K* was 2 (Fig 3C). When *K* was 3 to 5, the white-handed gibbon ancestry of each individual was almost evenly divided into 2 to 4 components, and the pileated gibbon ancestry was not changed across the different *K* values. Thus, we considered that the genetic makeup of the gibbon individuals was well represented by result when *K* = 2. Adult gibbons in Kaeng Krachan (*n* = 11 of 12) showed 0.995–0.996 for

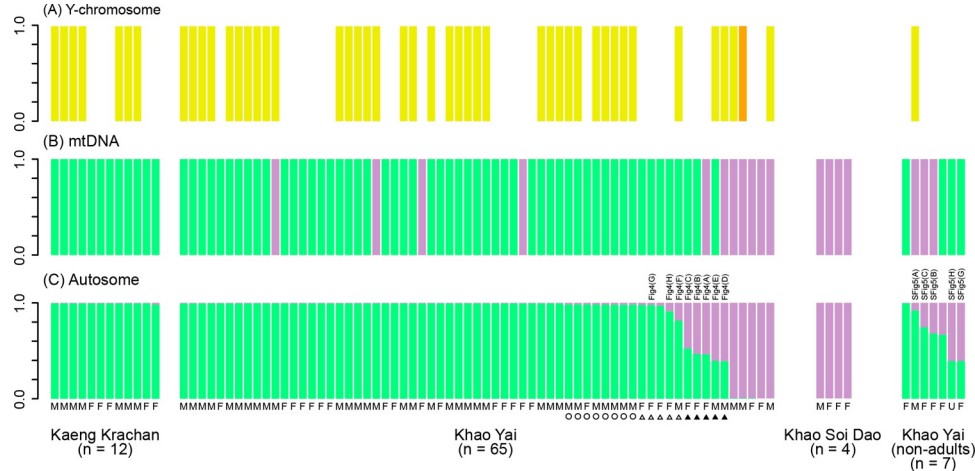

**Fig 3. White-handed and pileated gibbon ancestry of gibbons in Kaeng Krachan, Khao Yai and Khao Soi Dao.** (A) Y-chromosome. Yellow: haplogroup YA; orange: haplogroup YB. (B) mtDNA and (C) autosome. Green: white-handed gibbon ancestry; purple: pileated gibbon ancestry. Black triangles represent intermediate hybrid gibbons. Blank triangles represent hybrid gibbon with low levels of mixed ancestry. Blank circles represent "possible hybrid" gibbons with marginal mixed ancestry. M: male; F: female; U: sex unidentified.

white-handed ancestry and 0.004–0.005 for pileated ancestry, while the remaining individual showed marginally lower respective higher values of 0.983 for white-handed and 0.017 for pileated ancestry. On the other hand, adult gibbons in Khao Soi Dao ($n = 4$) showed 0.004 for white-handed and 0.996 for pileated ancestry.

In Khao Yai, levels of mixed ancestry were varied. Among 65 adult gibbons, 18 showed a mixed ancestry of white-handed (0.392 to 0.986) and pileated (0.014 to 0.608) gibbons (Fig 3C). Of those, five (three females, two males) showed intermediate values, approximately a half white-handed (0.392 to 0.527) and half pileated (0.473 to 0.608) ancestry, and five adult gibbons (four females, one male) showed stronger biased ancestry with low levels of pileated ancestry (0.022 to 0.185). We considered those ten gibbons as hybrid gibbons. The other eight gibbons (one female, seven males) with low levels of marginal pileated ancestry (0.014 to 0.018) but non-negligible levels of mixed ancestry (0.004 to 0.005), were considered "possible hybrid" gibbons. We based our decision on the fact that one pureblood white-handed gibbon in Kaeng Krachan showed pileated ancestry at 0.017, perhaps due to the small number of SNV loci used in the analysis and an incomplete segregation of SNV markers. Thus, a marginal value around 0.017 could be found in both hybrid and pureblood gibbons. Among seven non-adult gibbons observed in groups consisting intermediate hybrid adults, six showed mixed ancestry, with values of pileated ancestry ranging from 0.255 to 0.607. Another 42 adults and one non-adult gibbon showed quite low levels of pileated ancestry (0.004 to 0.005) at similar levels observed also among pureblood white-handed gibbons in Kaeng Krachan, and thus these individuals were identified as white-handed gibbons. Five adult gibbons showed large pileated ancestry (0.994 to 0.996) and almost no mixed ancestry of white-handed gibbons (0.004 to 0.006) at the same levels also observed among pureblood pileated gibbons in Khao Soi Dao, and thus these individuals were identified as pileated gibbons.

## Mitochondrial DNA and Y-chromosome ancestry

We found 32 haplotypes of HVR1 of mtDNA among 88 gibbons from the three study sites (S2 Table). Of those haplotypes, 21 newly found ones were deposited in DDBJ/EMBL/GenBank (Accession No. LC633853–LC633873). White-handed gibbons at Kaeng Krachan ($n = 12$) and pileated gibbons at Khao Soi Dao ($n = 4$) showed eight white-handed and three pileated typical mtDNA haplotypes. None of them were shared by Khao Yai gibbons (S1 Fig).

In Khao Yai, among ten adult hybrid gibbons, eight showed white-handed mtDNA haplotypes, and two showed the pileated mtDNA haplotype (Fig 3). All eight adult "possible hybrid" gibbons showed white-handed mtDNA haplotypes. Among 42 adult gibbons who were identified as white-handed gibbons based on the eight autosomal SNV loci, four shared a single pileated mtDNA haplotype (HKY11B) and the others showed white-handed mtDNA haplotypes. Conversely, among five adult gibbons who identified as pileated gibbons based on the eight autosomal SNVs, all showed only pileated mtDNA haplotypes. MtDNA haplotype of non-adult hybrid gibbons were identical to that of the adult hybrid female in the same group (HKY18B in AA1; HKY16A in AC4; HKY15A in AC3), suggesting mother-offspring relationships.

In the Y-chromosome, among Kaeng Krachan subjects, seven adult male white-handed gibbons showed only haplogroup YA. Among Khao Soi Dao subjects, we failed to amplify the Y-chromosome from the only pilated gibbon male in the sample. Among Khao Yai subjects, 26 adult male white-handed gibbons showed haplogroup YA, while three adult male pileated gibbons showed haplogroups YA ($n = 2$) and YB ($n = 1$), similar to previous results in captive gibbons [49, 50] (Fig 3). Because of the lack of Y-chromosome data for pileated gibbons at Khao Soi Dao, it was not possible to conclude whether or not the two haplogroups in pileated

gibbons at Khao Yai occurred because of recent introgression or a natural polymorphism (by incomplete lineage sorting or ancient introgression). The three adult hybrid males and the seven "possible hybrid" males showed haplogroup YA.

## Morphology and mixed ancestry

Of the five adult intermediate mixed ancestry gibbons, three were female and two were male. As expected for traits under strong genetic control, these individuals showed mixed/mosaic characteristics in pelage (Fig 4A–4E) as well as vocalization patterns (S2 Fig). The five hybrid gibbons showed fluffy hair (Fig 4A–4E), which is more characteristic of white-handed gibbons (S3 Fig) than the straighter hair of pileated gibbons (S4 Fig). The pelage coloration of two of the females was similar to one another (Fig 4A and 4B) while it was different in the third female. Their main body hair was creamy buff, similar to buff white-handed gibbons, while their ventral hair was black, similar to adult pileated gibbon females. Their head and face ring pelage color patterns were similar to those of young adult pileated females who still express white eyebrows. The hair on the hands and feet was white, which is typical for both species, but the white hair extended up until the middle of the forearms, which is further than in typical

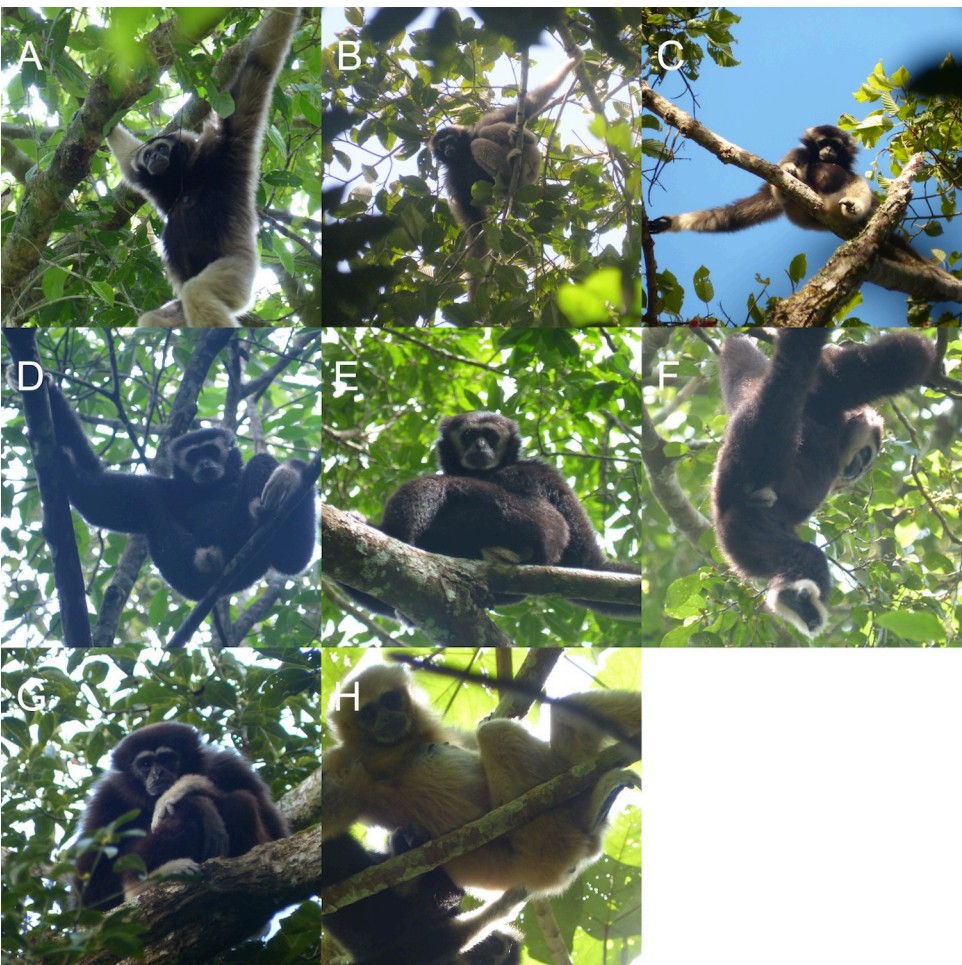

**Fig 4. External morphology of hybrid gibbons.** (A) Adult female hybrid of AA1. (B) Adult female hybrid of AB1. (C) Adult female hybrid of AC4. (D) Adult male hybrid of AA2. (E) Adult male hybrid of AA3. (F) Adult male hybrid of L. (G) Adult female hybrid of AC3. (H) Adult female hybrid of A. See Fig 3 for the degree of mixed ancestry.

white-handed gibbons, on which the white area usually reaches only around the wrists and ankles only, and more consistent with pileated gibbon's extent of white hair on the forearm. At the same time, the black hair on the chest and abdomen also extended to the internal part of proximal upper limbs, which is absent in typical adult pileated gibbon females. The third female's pelage pattern showed much more black hair on the entire body than the other two females, while her eyebrows, cap, hands and feet hairs resembled the white hair of the other two females (Fig 4C). The two adult hybrid males were similar to adult male pileated gibbons in that they expressed a white crown and white genital tuft, but were similar to white-handed gibbons in the expression of the white face ring and white hands and feet (Fig 4D and 4E).

Aligned with the SNVs analysis, other adult hybrid males and females with lower levels of mixed ancestry of pileated gibbon showed mosaic characteristics. For example, an adult male with 0.185 pileated ancestry showed a pelage color pattern of the black morph of white-handed gibbons but also showed a prominent white genital tuft not present in white-handed gibbons (Fig 4F). One adult female with 0.022 pileated ancestry also showed a color pattern similar to the black morph of white-handed gibbons, with a thinner white face ring (Fig 4G). One adult female with 0.088 pileated ancestry showed a typical buff white-handed gibbon pelage color (Fig 4H). Overall, three hybrid individuals with low levels of pileated ancestry were indistinguishable from typical white-handed gibbons, while another two hybrids with low levels of pileated ancestry showed atypical patterns. Among those, the extent of detected plieated ancestry and the presence/absence of atypical pelage color pattern were not related.

Pelage coloration of non-adult hybrid gibbons varied (S5 Fig). In addition to the six genetically analyzed, non-adult hybrid gibbons, there were several non-adult gibbons observed together with hybrid adults (Table 1). Many of them showed atypical characteristics for either species and thus appeared to be the offspring of hybrids. Some juveniles and adolescents

**Table 1. Gibbon groups with adult hybrid(s).**

| Group | Adult female | Adult male(s) | Offspring of the hybrid | Other non-adult members |
|---|---|---|---|---|
| AA1 | Hybrid† | *H. lar*† (primary) | 4 (3† and 1‡) | |
| | | "Possible hybrid"† (secondary) | | |
| AA2 | *H. lar*‡ | Hybrid† | 2‡ | |
| AA3 | *H. lar*‡ | Hybrid† (primary) | 1‡ | *H. lar*† subadult female |
| | | *H. lar*† (secondary) | | |
| AB1 | Hybrid† | *H. pileatus*‡ | 1‡ | |
| AC3 | Hybrid† (backcross with *H. lar*) | *H. pileatus*† | 3 (2† and 1‡) | |
| AC4 | Hybrid† | "Possible hybrid"§ | 2 (1† and 1‡) | |
| A | Hybrid§ (backcross with *H. lar*) | *H. lar*† (primary) | 2ɟ (both disperesd before 2010) | |
| | | *H. lar*† (secondary) | | |
| L† | *H. lar* | Hybrid† (backcross with *H. lar*) (primary) | | *H. lar*¥ subsdult female |
| | | | | *H. la*¥juvenile |
| | | *H. lar*† (secondary) | | *H. lar*¥ infant |
| R (status in 2010) | Hybrid§ (backcross with *H. lar*) | *H. lar*† | 1ɟ | |
| W (status in 2010) | Hybrid§ (backcross with *H. lar*) | *H. lar*† | 2ɟ | |

†: genetically and phenotypically confirmed

‡: phenotypically confirmed

§: mixed status was genetically confirmed, but not detected in phenotypes

ɟ: based on previous kinship analyses [16]

¥: genetically not tested.

showed a color pattern similar to pileated gibbons: whitish-grey with black patches on the abdominal area and the top of the head (S5 Fig).

## Vocalization and mixed ancestry

As previously reported, both male and female intermediate hybrid gibbons sang phrases different from those of pure white-handed or pileated gibbons [15, 32, 54]. This was especially detectable in the female great call–male coda sequence (S4 Fig) but also other phrases. There were atypical notes among hybrid gibbons, including non-adult individuals with various levels of mixed ancestry.

## Group composition of white-handed gibbons, pileated gibbons and their hybrids

Both, adult hybrid females and males formed groups (Table 1). The intermediate hybrid females formed a pair or a multi-male/one-female trio with varying male types (groups AA1, AB1 and AC4). Intermediate hybrid males formed a pair or a multi-male/one female trio with a white-handed female and a white-handed secondary male (groups AA2 and AA3). Although we found no intermediate hybrid male paired with a pileated female, this result was probably an outcome of our small sample size and white-handed gibbons mainly occupied the study area. As previously explained (i.e., see sections of external morphology), all five adult intermediate hybrid gibbons were assumed to have reproduced because many non-adult gibbons in their groups equally showed external morphologies inconsistent with either pureblood-species (S5 Fig).

The other five hybrid gibbons with lower levels of pileated ancestry also formed groups (Table 1). Four of them formed groups with white-handed gibbons, and the last formed a group with a pileated gibbon. The presence of putative offspring in several of the groups suggested that at least four of the five gibbons had reproduced.

We found five groups formed with two different species (Table 2), of which four included individuals who could not be genotyped because of missing fecal samples. However, pelage color and vocalization of non-genotyped groups suggested no or minimal mixed ancestry. Of these groups, only one was formed by a pair with one pileated female and one white-handed male (group AB3). Three were multi-male groups: each with a white-handed and a pileated gibbon male paired with a white-handed gibbon female (groups AB2, AB5 and AD1). The remaining group was composed of a white-handed and a pileated gibbon female with two white-handed gibbon males (group AA10). In those five groups, no hybrid offspring were detected (Table 2).

**Table 2. Gibbon groups made up of heterospecific members.**

| Group | Adult female | Adult male(s) | Non-adult members |
|-------|--------------|---------------|-------------------|
| AA10 | *H. lar* | *H. lar* (primary) | None |
| | *H. pileatus* | *H. lar* (secondary) | |
| AB2 | *H. lar* | *H. lar* (primary?) | *H. lar* juvenile |
| | | *H. pileatus* (secondary?) | |
| AB3 | *H. pileatus* | *H. lar* | *H. pileatus* subadult female |
| AB5 | *H. lar* | *H. lar* (primary?) | *H. lar* adolescent, *H. lar* juvenile |
| | | *H. pileatus* (secondary?) | |
| AD1 | *H. lar* | *H. lar* (primary?) | *H. lar* subadult male (?), *H. lar* adolescent, *H. lar* juvenile |
| | | *H. pileatus* (secondary?) | |

## Discussion

Although, we previously reported the presence of phenotypical white-handed gibbons with a pileated-type mtDNA among Khao Yai gibbons [34], we were unable to address the levels of admixture because of a lack of autosomal DNA information at that time. In this study, we have identified hybrid gibbons with various levels of mixed ancestry by using autosomal SNV markers and mtDNA. Based on the estimated admixture levels, hybrid gibbons could represent a spectrum of first filial to plural backcross generations. Interestingly, four adult individuals (two males, two females) with white-handed typical phenotypic appearance but a pileated mtDNA haplotype (HKY11B) did not show mixed ancestries in their nuclear genomes, as represented by eight SNV markers (Fig 3). Thus, hybridization and introgression between white-handed and pileated gibbons in Khao Yai appears to be a rather non-recent event that has resulted from a series of repeated admixture events along multiple generations. The small number of SNV loci used in the present study together with the use of fecal samples restricted further investigations such as the number of generations passed after the initial hybridization event between the two species. Recent technological innovations in the use of fecal samples of wild primates for genome-wide analysis [55, 56] may help to obtain more information on the hybridization between the two gibbon species in Khao Yai.

We did not detect apparent survival or reproductive disadvantages in hybrid gibbons in the Khao Yai population and both male and female intermediate hybrids formed pair bonds with either white-handed or pileated gibbons. Furthermore, intermediate hybrid gibbons were with one to four non-adult individuals (mean = 2) in their groups with species-atypical phenotypes consistent with a genetically confirmed mixed ancestry for some individuals. The number of non-adult individuals (presumably offspring) observed in groups was within the range observed for white-handed gibbons (0–4) and pileated gibbons (1–3) in Khao Yai, Kaeng Krachan and Khao Soi Dao (the present study and [25]). These findings support the preliminary conclusion that the reproductive success of hybrids and parental species are not significantly different from one another. Moreover, the presence of subadult offspring suggests the stable position of the hybrids in the groups. However, only a molecular parentage analysis of paternity will be able to assess if hybrid males reproduce as effectively as males of parental species and likewise long-term molecular monitoring will be needed to assess reproducve rates in hybrid and parental species groups at Khao Yai.

Overall, in Khao Yai, a long history of hybridization and introgression between white-handed and pileated gibbons is strongly supported by our autosomal SNV analysis. A moderate tempo and mode of hybridization and introgression coupled with no apparent disadvantages in hybrids could have established the introgression pattern seen between the two species today. After the divergence of white-handed and pileated gibbons around 3–4 million years ago [35, 44, 57, 58], it is readily conceivable that hybridization and introgression occurred repeatedly where contact zones had formed along the species' distribution boundaries. Together with behavioral data, genome-wide analyses focusing on Khao Yai gibbons, can shed further light on introgression and the evolutionary consequences of hybridization in small apes.

## Supporting information

**S1 Fig. Median-joining network of hypervariable region 1 of mtDNA.** Short bars on branches indicate the number of substitutions between nodes. Node size reflects the number of each haplotype observed among 88 gibbons. HKY: 21 haplotypes observed at Khao Yai; HKK: 8 haplotypes observed at Kaneg Krachan; HKSD: 3 haplotypes observed at Khao Soi

Dao.
(TIF)

**S2 Fig. Sonograms of female great call–male coda sequence of white-handed, pileated and hybrid gibbons.** (A) A white-handed gibbon pair at Kaeng Krachan. (B) A pileated gibbon pair at Khao Soi Dao. (C) A hybrid female and a white-handed male pair at Khao Yai. (D) A white-handed female and a hybrid male pair at Khao Yai.
(TIF)

**S3 Fig. Pelage color of white-handed gibbons.** (A) Adult male (black morphotype). (B) Adult female (buff morphotype) and infant (buff morphotype). (C) Adult female (black morphotype) and infant (black morphotype). (D) Juvenile (buff morphotype).
(TIF)

**S4 Fig. Pelage color of pileated gibbons.** (A) Adult male. (B) Adult female. (C) Adult female and infant. (D) Adolescent.
(TIF)

**S5 Fig. Pelage color of non-adult hybrid gibbons.** (A) Subadult male of AA1. (B) Adolescent female of AA1. (C) Juvenile female of AA1 (A–C: putative mother = intermediate hybrid). (D) Subadult male of AA2. (E) Juvenile of AA2 (D–E: putative father = intermediate hybrid). (F) Juvenile of AB1 (putative mother = intermediate hybrid). (G) Adolescent female of AC3. (H) Juvenile of AC3. (I) Infant of AC3 (G–I: putative mother = hybrid with a low level of mixed ancestry).
(TIF)

**S1 Table. Status of SNV sites and primer–probe information.**
(XLSX)

**S2 Table. Samples and genotypes.**
(XLSX)

## Acknowledgments

We thank Khao Yai National Park, Kaeng Krachan National Park and Khao Soi Dao Wildlife Sanctuary, the Department of National Parks, Wildlife and Plant Conservation, and the National Research Council of Thailand for giving us permissions to conduct the study. We thank Mr. Krissada Homsud and Mr. Kanchit Srinoppawan, the superintendents of Khao Yai National Park, Mr. Kamol Nuanyai, the superintendent of Kaeng Krachan National Park, and Mr. Sidtichai Bunphot, the superintendent of Khao Soi Dao Wildlife Sanctuary, for their kind support during our staying in the area. We also appreciate Mr. Phanakorn Kraomklang (Khao Yai National Park), Mr. Paitoon Chamted and Mr. Wisoot Supong (Kaeng Krachang National Park), and Mr. Suriya Laapwiseet (Khao Soi Dao Wildlife Sanctuary) and many other park officials for their kind help in conducting the fieldwork. We appreciate Drs. Warren Brockelman, Norberto Asensio, Chalita Kongrit, and Intanon Kolasatsanee for their kind suggestions for the field survey. We also thank members of the Primate Research Unit, Chulalongkorn University, and Dr. Wannapa Ishida for their kind help throughout the study.

## Author Contributions

**Conceptualization:** Kazunari Matsudaira.

**Data curation:** Kazunari Matsudaira.

**Formal analysis:** Kazunari Matsudaira.

**Funding acquisition:** Kazunari Matsudaira.

**Investigation:** Kazunari Matsudaira.

**Project administration:** Kazunari Matsudaira.

**Resources:** Kazunari Matsudaira, Ulrich H. Reichard, Takafumi Ishida, Suchinda Malaivijitnond.

**Supervision:** Suchinda Malaivijitnond.

**Visualization:** Kazunari Matsudaira.

**Writing – original draft:** Kazunari Matsudaira.

**Writing – review & editing:** Ulrich H. Reichard, Takafumi Ishida, Suchinda Malaivijitnond.

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
