## [Decision Letter · Decision Letter 0]

9 Oct 2021

PONE-D-21-19052Introgression and mating patterns between white-handed gibbons (Hylobates lar) and pileated gibbons (Hylobates pileatus) in a natural hybrid zonePLOS ONE

Dear Dr. Matsudaira,

Thank you for submitting your manuscript to PLOS ONE. After careful consideration, we feel that it has merit but does not fully meet PLOS ONE’s publication criteria as it currently stands. Therefore, we invite you to submit a revised version of the manuscript that addresses the points raised during the review process.

Both of the reviewers agree on that the limited number of genetic markers in this study cannot provide enough resolution to clearly distinguish F1s and backcrossed individuals or to evaluate the degree of gene flow. I have to say that the conclusions are not fully justified by the data in the current manuscript. However, I agree with the reviewer #1 on that this manuscript is publishable if the authors can revise the manuscript and tone down the conclusion. Therefore, my decision is “Major Revision”.

There was a disagreement between the reviewers regarding the limited number of available SNPs or genetic markers. Considering that obtaining larger sample size than the current data is difficult, I am more inclined to agree with the reviewer #1. But as reviewer #2 suggested, please explain the difficulty of obtaining samples and justify the reason why the only the limited sample size and the limited number of genetic markers were used.

There was some unclear points on how the genetic markers were selected and which individuals are included in this study. Please clarify these points on the revised manuscript because these information is essential to make the manuscript technically sound.

Also please address the other points raised by the reviewers than I mentioned below.

We look forward to receiving your revised manuscript.

Kind regards,

Jun Gojobori

Academic Editor

PLOS ONE

Journal Requirements:

2. We note that Figure 1 &  2 in your submission contain [map/satellite] images which may be copyrighted. All PLOS content is published under the Creative Commons Attribution License (CC BY 4.0), which means that the manuscript, images, and Supporting Information files will be freely available online, and any third party is permitted to access, download, copy, distribute, and use these materials in any way, even commercially, with proper attribution. For these reasons, we cannot publish previously copyrighted maps or satellite images created using proprietary data, such as Google software (Google Maps, Street View, and Earth). For more information, see our copyright guidelines: http://journals.plos.org/plosone/s/licenses-and-copyright.

a. You may seek permission from the original copyright holder of Figure(s) [#] to publish the content specifically under the CC BY 4.0 license.

We recommend that you contact the original copyright holder with the Content Permission Form (http://journals.plos.org/plosone/s/file?id=7c09/content-permission-form.pdf) and the following text: “I request permission for the open-access journal PLOS ONE to publish XXX under the Creative Commons Attribution License (CCAL) CC BY 4.0 (http://creativecommons.org/licenses/by/4.0/). Please be aware that this license allows unrestricted use and distribution, even commercially, by third parties. Please reply and provide explicit written permission to publish XXX under a CC BY license and complete the attached form.”

b. If you are unable to obtain permission from the original copyright holder to publish these figures under the CC BY 4.0 license or if the copyright holder’s requirements are incompatible with the CC BY 4.0 license, please either i) remove the figure or ii) supply a replacement figure that complies with the CC BY 4.0 license. Please check copyright information on all replacement figures and update the figure caption with source information. If applicable, please specify in the figure caption text when a figure is similar but not identical to the original image and is therefore for illustrative purposes only. The following resources for replacing copyrighted map figures may be helpful:

USGS National Map Viewer (public domain): http://viewer.nationalmap.gov/viewer/ The Gateway to Astronaut Photography of Earth (public domain): http://eol.jsc.nasa.gov/sseop/clickmap/ Maps at the CIA (public domain): https://www.cia.gov/library/publications/the-world-factbook/index.html and https://www.cia.gov/library/publications/cia-maps-publications/index.html NASA Earth Observatory (public domain): http://earthobservatory.nasa.gov/ Landsat: http://landsat.visibleearth.nasa.gov/ USGS EROS (Earth Resources Observatory and Science (EROS) Center) (public domain): http://eros.usgs.gov/# Natural Earth (public domain): http://www.naturalearthdata.com/

Reviewers' comments:

Reviewer's Responses to Questions

**Comments to the Author**

1. Is the manuscript technically sound, and do the data support the conclusions?

Reviewer #1: Partly

Reviewer #2: Partly

2. Has the statistical analysis been performed appropriately and rigorously? 

Reviewer #1: Yes

Reviewer #2: Yes

3. Have the authors made all data underlying the findings in their manuscript fully available?

Reviewer #1: Yes

Reviewer #2: Yes

4. Is the manuscript presented in an intelligible fashion and written in standard English?

Reviewer #1: Yes

Reviewer #2: Yes

5. Review Comments to the Author

Reviewer #1: Matsudaira and colleges present genetic and phenotypic evidence for hybridisation of white-handed and pileated gibbons in a hybrid zone.

Being unfamiliar with gibbon phenotypic diversity I restrict my comments to the genetic analysis which is within my area of expertise.

The authors present a genetic analysis, using 8 autosomal, MT and Y markers. This appears a novel investigation of a recent and/or ongoing hybridisation of these two species, and a counterpart to the investigation of older introgression described in Matsudaira and Ishida Heredity volume 127, pages312–322 (2021).

As such, I would be pleased if ultimately this paper was published in PLoS One. There is however a question as to how strong the interpretation of the genetic data is.

I certainly would agree with the authors that even a small number of suitably chosen loci can readily differentiate between two diverged lineages, and that these would also provide some information as to introgression/hybrid formation and/or a cline of ancestry. I disagree that there is enough resolution to readily differentiate between F1/F2 or various generations of repeat backcrosses. Given the small number of autosomal markers (less than the number of chromosomes), and the associated uncertainty in estimated ancestry components - even with only K = 2 - many different population processes could result in very similar arrays of ancestry that the authors uncover within the hybrid zone. Thus, I think it is fair to conclude that the authors uncover genetic evidence for recent hybridisation (Recent being relative to the related findings in Matsudaira and Ishida 2021). I do not think that one can say, as the authors do in the Abstract "Four phenotypically white-handed gibbons showed introgressed pileated type mtDNA, but the autosomal SNV admixture was not detected" and therefore ... "some genomic portions have introgressed into the gene pool of the counterpart species". In my opinion, this is too strong an interpretation of their results. I would be much happier if statements of this kind came with a suitable disclaimer. For example: "These gibbons were not only F1 hybrids, but also ones with plural backcross generations" (Lines 405:406). Much better could be, "Our data suggests that these gibbons could be a mixture of F1 hybrids, and perhaps ones with plural backcross generations, but the limited resolution enabled by small number of marker loci cannot rule out other scenarios acting on shorter timescales, such as a mixture of F1/F2 individuals". Of course how the authors want to address this point is up to them. Also, please note this is but one example, and my comment should be interpreted as having the authors rethink all statements on genetic ancestry that are definitive in tone such as Lines 428-429.

Minor points:

Data availability. I searched all three databases and could not find the listed accessions. Can the authors please confirm the availability of data.

Selection of autosomal SNVs: please expand on the relevant methods. How many DNA sequences from GenBank were initially selected?; how many independent fixed differences did you find? How many additional loci came from Perelman et al?; In essence - the reader should have some clarity about how you came to only 8 SNVs.

Lines 70-71: citation for disadvantages of hybridisation.

Reviewer #2: In this manuscript, “Introgression and mating patterns between white-handed gibbons (Hylobates lar) and pileated gibbons (Hylobates pileatus) in a natural hybrid zone”, the authors seek to uncover the degree of introgression and mating patterns between two endangered gibbon species, white-handed and pileated gibbons. They genotyped 88 individuals including 72 individuals from the hybrid zone, 12 purebred white-handed and 4 purebred pileated gibbons, with 8 autosomal SNPs. They also determined 632 bp-sequence on mtDNA and one 2bp-indel on Y chromosome for most of the individuals.

However, as I explained below, I don’t think it’s possible to measure the extent of introgression genetically with this data because of the small number of SNPs and the small number of the purebred samples. Rather, the importance of this study is in proposing potential markers to distinguish the two species and in accumulating genotype information of these endangered species. I recommend to reorganize this manuscript in that way.

1. The major concern is the small number of SNPs analyzed in this manuscript (8 SNPs) and the small number of the purebred individuals (12 white-handed and 4 pileated gibbons). It is hard to say that these SNPs are fixed differences between these species. The authors chose the 8 autosomal SNPs from the previous works of other groups; 5 SNPs from Chan et al. (2013) and 3 SNPs from Perelman et al. (2011). But the numbers of the animals used in the papers are also very small. In Chan et al., 11 white-handed and 4 pileated gibbons were used, and in Perelman et al., only one white-handed gibbon is used. It means, the genotypes of the 8 autosomal SNPs are available only for 13 or 23 white-handed gibbons and 4 or 8 pileated gibbons in total. It is likely that the 8SNPs are not fixed differences between the species but are segregating at high frequency within each species. In that case, it’s not possible to distinguish the two species with these SNPs.

2. In lines 269-270 and in Fig 3., the authors mentioned “the threshold of backcross and pure white-handed gibbons”. How was it determined? Also, how the criteria between F1 and backcross was determined (Fig3)? I guess it came from the identification of species base on vocalization and morphology. If so, it means that in this manuscript, the species identification is not based on genetic data but based simply on the vocalization and morphology. The aim of this manuscript is written as analyzing the samples genetically to evaluate the extent of introgression, but the actual analyses are done the other way round. I think the authors’ true interest is in finding a set of potential markers to distinguish species for future studies and to confirm hybrids which are identified from vocalization and morphology. I do understand the importance of finding the markers for endangered species, for which it’s difficult (/not possible) to get good amount of DNA samples. But the manuscript needs to be modified to make the aim clear.

3. The sample information is not clear enough. The authors collected 129 fecal samples in Khao Yai, 71 fecal samples in Kaeng Krachan and 9 fecal samples on Khao Soi Dao (lines179-181). They also genotyped DNA samples previously collected in Khao Yai (lines 193-194). Then suddenly declared “Overall, 93 DNA samples of 88 individuals were genotyped” without any explanations. Please provide the full sample list with sample ID, sampling location, sex, age category, species identified by vocalization and pelage coloration, and information from genetic data. If the data includes the samples which were used in previous studies, please explain it with the same sample IDs as in the previous works.

4. I think the authors should explain about the endangered status of the two species more, if that is the reason why the amount of genetic data is very limited.

5. The authors wrote “The electropherograms of HVR1 were visualized by MEGA6 " (line 247). Is it a mistake? I couldn’t find electropherograms in this manuscript.

6. PLOS authors have the option to publish the peer review history of their article (what does this mean?). If published, this will include your full peer review and any attached files.

Reviewer #1: No

Reviewer #2: No

---

## [Author Response · Author response to Decision Letter 0]

15 Nov 2021

Dear Dr. Gojobori,

Thank you very much for evaluating the manuscript entitled “Introgression and mating patterns between white-handed gibbons (Hylobates lar) and pileated gibbons (Hylobates pileatus) in a natural hybrid zone”. We revised the manuscript following your comments and the two reviewers. The major point is that we toned down the conclusion on the level of admixture of each individual due to the limited number of SNV markers. Please find the details of revisions in the replies below.

In addition, submitting the agreements from copyright holders of the map information for Fig1 and Fig2 were requested. However, we recreated the maps by using only the data provided under the CC BY-SA license and describe the information in the figure legends appropriately. Therefore, we believe the agreements are not needed for these cases.

Comments from the editor

Thank you for submitting your manuscript to PLOS ONE. After careful consideration, we feel that it has merit but does not fully meet PLOS ONE’s publication criteria as it currently stands. Therefore, we invite you to submit a revised version of the manuscript that addresses the points raised during the review process.

Both of the reviewers agree on that the limited number of genetic markers in this study cannot provide enough resolution to clearly distinguish F1s and backcrossed individuals or to evaluate the degree of gene flow. I have to say that the conclusions are not fully justified by the data in the current manuscript. However, I agree with the reviewer #1 on that this manuscript is publishable if the authors can revise the manuscript and tone down the conclusion. Therefore, my decision is “Major Revision”.

There was a disagreement between the reviewers regarding the limited number of available SNPs or genetic markers. Considering that obtaining larger sample size than the current data is difficult, I am more inclined to agree with the reviewer #1. But as reviewer #2 suggested, please explain the difficulty of obtaining samples and justify the reason why the only the limited sample size and the limited number of genetic markers were used.

Answer: Thank you very much for your kind understanding on the difficulties in obtaining large number of both samples and appropriate markers. We revised the manuscript and added the explanation of difficulties in the sampling from unhabituated gibbons (line 187-196) and finding markers (line 211-234). We also agree with you and reviewer #1 at the point of toning down the conclusion because of the difficulties in distinguishing first filial (F1) and backcrosses. Therefore, we avoided concluding the individuals as F1 by replacing the sentences.

There was some unclear points on how the genetic markers were selected and which individuals are included in this study. Please clarify these points on the revised manuscript because these information is essential to make the manuscript technically sound.

Answer: We revised and added detailed information about the finding of markers (line 211-234).

Also please address the other points raised by the reviewers than I mentioned below.

Answer: We revised accordingly. Please find replies below for the details.

Review Comments to the Author

Reviewer #1: Matsudaira and colleges present genetic and phenotypic evidence for hybridisation of white-handed and pileated gibbons in a hybrid zone.

Being unfamiliar with gibbon phenotypic diversity I restrict my comments to the genetic analysis which is within my area of expertise.

The authors present a genetic analysis, using 8 autosomal, MT and Y markers. This appears a novel investigation of a recent and/or ongoing hybridisation of these two species, and a counterpart to the investigation of older introgression described in Matsudaira and Ishida Heredity volume 127, pages312–322 (2021).

As such, I would be pleased if ultimately this paper was published in PLoS One. There is however a question as to how strong the interpretation of the genetic data is.

I certainly would agree with the authors that even a small number of suitably chosen loci can readily differentiate between two diverged lineages, and that these would also provide some information as to introgression/hybrid formation and/or a cline of ancestry. I disagree that there is enough resolution to readily differentiate between F1/F2 or various generations of repeat backcrosses. Given the small number of autosomal markers (less than the number of chromosomes), and the associated uncertainty in estimated ancestry components- even with only K = 2 - many different population processes could result in very similar arrays of ancestry that the authors uncover within the hybrid zone. Thus, I think it is fair to conclude that the authors uncover genetic evidence for recent hybridisation (Recent being relativeto the related findings in Matsudaira and Ishida 2021). I do not think that one can say, as the authors do in the Abstract "Four phenotypically white-handed gibbons showed introgressed pileated type mtDNA, but the autosomal SNV admixture was not detected" and therefore ... "some genomic portions have introgressed into the gene pool of the counterpart species". In my opinion, this is too strong an interpretation of their results. I would be much happier if statements of this kind came with a suitable disclaimer. For example: "These gibbons were not only F1 hybrids, but also ones with plural backcross generations" (Lines 405:406). Much better could be, "Our data suggests that these gibbons could be a mixture of F1 hybrids, and perhaps ones with plural backcross generations, but the limited resolution enabled by small number of marker loci cannot rule out other scenarios acting on shorter timescales, such as a mixture ofF1/F2 individuals". Of course how the authors want to address this point is up to them. Also, please note this is but one example, and my comment should be interpreted as having the authors rethink all statements on genetic ancestry that are definitive in tone such as Lines 428-429.

Answer: Thank you very much for your kind comments on improving the manuscript. We agree with your comments that the small number of SNV markers used in the study has limitations in determining the generation of hybrids. We revised the manuscript and avoid the use of the term, F1, for the individuals who showed intermediate admixture levels. We also toned down the conclusion following the comments.

Minor points:

Data availability. I searched all three databases and could not find the listed accessions. Can the authors please confirm the availability of data.

Answer: The sequences were registered and thus the accession numbers were assigned by DDBJ. Opening the sequences for the public is pending. We will open the data as soon as the manuscript is accepted.

Selection of autosomal SNVs: please expand on the relevant methods. How many DNA sequences from GenBank were initially selected?; how many independent fixed differences did you find? How many additional loci came from Perelman et al?; In essence – the reader should have some clarity about how you came to only 8 SNVs.

Answer: We followed the comments and added detailed explanations for the screening process of SNVs (line 211-234).

Lines 70-71: citation for disadvantages of hybridisation.

Answer: We added the citation and an example of disadvantage (line 67-70).

Reviewer #2: In this manuscript, “Introgression and mating patterns between white-handed gibbons (Hylobates lar) and pileated gibbons (Hylobates pileatus) in a natural hybrid zone”, the authors seek to uncover the degree of introgression and mating patterns between two endangered gibbon species, white-handed and pileated gibbons. They genotyped 88 individuals including 72 individuals from the hybrid zone, 12 purebred white-handed and 4 purebred pileated gibbons, with 8 autosomal SNPs. They also determined 632 bp-sequence on mtDNA and one 2bp-indel on Y chromosome for most of the individuals.

However, as I explained below, I don’t think it’s possible to measure the extent of introgression genetically with this data because of the small number of SNPs and the small number of the purebred samples. Rather, the importance of this study is in proposing potential markers to distinguish the two species and in accumulating genotype information of these endangered species. I recommend to reorganize this manuscript in that way.

Answer: Thank you very much for your kind comments on the manuscript. We agree that our previous conclusion was overstated against the small number of SNV markers used in the study and thus some ambiguity in the resolution of evaluating the admixture level remained. However, as we can see in the results, STRUCTURE analysis successfully distinguished pureblood white-handed gibbons and pileated gibbons without prior information of the species status in these samples (i.e. STRUCTURE did not know which samples were of white-handed gibbons and which samples were of pileated gibbons). Therefore, the finding of the presence of hybrid individuals with various levels of admixture level is robust. Following the comments from the editor, Dr. Gojobori, and reviewer #1, we toned down the conclusion. We believe our revisions made the manuscript scientifically feasible.

1. The major concern is the small number of SNPs analyzed in this manuscript (8 SNPs) and the small number of the purebred individuals (12 white-handed and 4 pileated gibbons). It is hard to say that these SNPs are fixed differences between these species. The authors chose the 8 autosomal SNPs from the previous works of other groups; 5 SNPs from Chan et al. (2013) and 3 SNPs from Perelman et al. (2011). But the numbers of the animals used in the papers are also very small. In Chan et al., 11 white-handed and 4 pileated gibbons were used, and in Perelman et al., only one white-handed gibbon is used. It means, the genotypes of the 8 autosomal SNPs are available only for 13 or 23 white-handed gibbons and 4 or 8 pileated gibbons in total. It is likely that the 8 SNPs are not fixed differences between the species but are segregating at high frequency within each species. In that case, it’s not possible to distinguish the two species with these SNPs.

Answer: Technically saying, genetic markers used in STRUCTURE analysis do not have to be fixed in K species/populations. STRUCTURE assumes the presence of hidden K clusters among the samples, where each cluster has a particular frequency of each SNV. In this process, STRUCTURE does not know which sample comes from which cluster a priori. Therefore, if the number of SNV markers and the number of pureblood white-handed gibbons and pureblood pileated gibbons were too small to distinguish the two species, the membership coefficient shown by STRUCTURE should have not represented the ancestry of the two species. Fortunately, as we can see in the results, 12 gibbons from Kaeng Krachan and 4 gibbons from Khao Soi Dao showed a quite small amount of mixed ancestry which can be considered the noise usually observed in STRUCTURE analysis, and thus (it is safe to say, that) the number of SNV markers and number of pureblood gibbons used in the study was enough for detecting hybrid gibbons with admixed ancestry.

On the other hand, we agree that the number of the SNV markers was not enough to clearly distinguish first filial, second filial and several backcross generations as suggested by the editor and reviewer #1, we revised this point and toned down the conclusion.

2. In lines 269-270 and in Fig 3., the authors mentioned “the threshold of backcross and pure white-handed gibbons”. How was it determined? Also, how the criteria between F1 and backcross was determined (Fig3)? I guess it came from the identification of species base on vocalization and morphology. If so, it means that in this manuscript, the species identification is not based on genetic data but based simply on the vocalization and morphology. The aim of this manuscript is written as analyzing the samples genetically to evaluate the extent of introgression, but the actual analyses are done the other way round. I think the authors’ true interest is in finding a set of potential markers to distinguish species for future studies and to confirm hybrids which are identified from vocalization and morphology. I do understand the importance of finding the markers for endangered species, for which it’s difficult (/not possible) to get good amount of DNA samples. But the manuscript needs to be modified to make the aim clear.

Answer: The threshold of backcross and pureblood white-handed gibbons was determined based on the observed admixed ancestry values in 12 pureblood white-handed gibbons from Kaeng Krachan and 4 pureblood pileated gibbons from Khao Soi Dao. Because of the characteristic of STRUCTURE analysis, a small amount of mixed ancestry could be detected in pureblood samples. Therefore, the maximum admixed ancestry value 0.017 observed in one Kaeng Krachan white-handed gibbon was defined as the threshold. We added an explanation on this point (line 279-287). The threshold between the F1 and backcross was determined arbitrary and thus was inappropriate. Therefore, we revised and removed the distinction between F1 and backcross, and considered all the individuals as mere hybrids.

3. The sample information is not clear enough. The authors collected 129 fecal samples in Khao Yai, 71 fecal samples in Kaeng Krachanand 9 fecal samples on Khao Soi Dao (lines179-181). They also genotyped DNA samples previously collected in Khao Yai (lines 193-194). Then suddenly declared “Overall, 93 DNA samples of 88 individuals were genotyped” without any explanations. Please provide the full sample list with sample ID, sampling location, sex, age category, species identified by vocalization and pelage coloration, and information from genetic data. If the data includes the samples

which were used in previous studies, please explain it with the same sample IDs as in the previous works.

Answer: We restructured the section and made the number of individuals used in the analysis clear (line 187-196). We also added the information of individuals, fecal samples, genotypes etc. as Supplementary Table 2 following the comments from reviewer #2. We hope the additional information will help to understand the details of the subjects.

4. I think the authors should explain about the endangered status of the two species more, if that is the reason why the amount of genetic data is very limited.

Answer: The limitation of sample number is derived from the difficulty in collecting fecal samples from unhabituated gibbons (it took many days to collect fecal samples from each individual). Also, the limitation of the genetic markers was mainly due to the financial issue for designing primers and probes rather than the available sequences from the database.

5. The authors wrote “The electropherograms of HVR1 were visualized by MEGA6 " (line 247). Is it a mistake? I couldn’t find electropherograms in this manuscript.

Answer: Electropherogram is raw data of electrophoresis by a Sanger sequencing and thus usually not informative to be shown in the manuscript without particular reason (e.g. to emphasize the appearance of heterozygous sites in the raw data). It is possible to show an example of one sequencing reaction, but we don’t think it is essential for this manuscript. To avoid misunderstanding what the electropherogram is we revised the sentence and add an explanation (line 265-266).

---

## [Decision Letter · Decision Letter 1]

20 Jan 2022

PONE-D-21-19052R1

Introgression and mating patterns between white-handed gibbons (Hylobates lar) and pileated gibbons (Hylobates pileatus) in a natural hybrid zone

PLOS ONE

Dear Dr. Matsudaira,

Thank you for submitting your manuscript to PLOS ONE. After careful consideration, we feel that it has merit but does not fully meet PLOS ONE’s publication criteria as it currently stands. Therefore, we invite you to submit a revised version of the manuscript that addresses the points raised during the review process.

I generally agree with the two reviewers and I think they made a reasonable suggestions. I ask the authors to address the comments raised by the reviewers before this manuscript is published.

We look forward to receiving your revised manuscript.

Kind regards,

Jun Gojobori

Academic Editor

PLOS ONE

Journal Requirements:

Reviewers' comments:

Reviewer's Responses to Questions

**Comments to the Author**

1. If the authors have adequately addressed your comments raised in a previous round of review and you feel that this manuscript is now acceptable for publication, you may indicate that here to bypass the “Comments to the Author” section, enter your conflict of interest statement in the “Confidential to Editor” section, and submit your "Accept" recommendation.

Reviewer #3: (No Response)

Reviewer #4: (No Response)

2. Is the manuscript technically sound, and do the data support the conclusions?

Reviewer #3: Yes

Reviewer #4: Yes

3. Has the statistical analysis been performed appropriately and rigorously? 

Reviewer #3: N/A

Reviewer #4: Yes

4. Have the authors made all data underlying the findings in their manuscript fully available?

Reviewer #3: Yes

Reviewer #4: Yes

5. Is the manuscript presented in an intelligible fashion and written in standard English?

Reviewer #3: Yes

Reviewer #4: Yes

6. Review Comments to the Author

Reviewer #3: I agree with both previous reviewers that the limited amount of data presented here should not be over-interpreted. In particular, I agree with the statement 1 of reviewer #2, the previously described SNPs might not represent fixed differences with the small number of individuals, especially since ancient admixture events were likely part of their history.

First, the observation in the STRUCTURE analysis could be explained by genetic clines or simply segregating variation rather than recent hybridization. There is one thing I would recommend in terms of analysis, which is a STRUCTURE analysis with higher k (3, 4, 5) to see if the supposed signature of mixing is not just a component restricted to individuals from this area.

Second, the threshold for assigning “hybrid ancestry” seems somewhat low and arbitrary given the small number of loci. Inspecting Fig. 3, there are 5-7 likely hybrids with substantial ancestry from both species (if one assumes these loci are really fixed), while I would be more doubtful about the others. The results on morphology and pelage are lending good support to the hypothesis on observed hybrids, especially in the case of those with clear mixed ancestries. I would like to know if the “five hybrid individuals indistinguishable from typical white-handed gibbons” had a lower assignment of admixed ancestry than the three with atypical patterns (L373).

Third, the discordance of mtDNA nuclear DNA might be pointed out. There are a few individuals with a haplotype assigned to H. pileatus which, in this restricted dataset, have nuclear ancestry only assigned to H. lar. This shows the difficulty in determining ancestries with such few markers. Compare that to humans and Neanderthals, where ancient Neanderthal admixture in modern humans has not been determined by obtaining several mtDNA sequences, nor by retrieving a small amount of nuclear DNA (Green et al., Nature 2006), but only by shallow genome sequencing (Green et al., Science 2010). I agree that it is difficult to obtain information from wild-living primates for this kind of studies, but it is a problem that the confidence in observations must be low when using such a small number of individuals for the reference populations and a very small number of genetic markers. In that sense, I would also add “putative” to any statement about admixture, from the abstract on throughout the paper. I.e. the phrasing should be even more cautious when referring to the genetic results.

Finally, technological innovations may likely offer opportunities to obtain much more information from fecal samples of wild primates, as has been shown in chimpanzees with sequencing after capture (see Fontsere et al., Molecular Ecology 2021). Considering the very fine-grained picture that were obtained in humans and other primate species after sequencing at least substantial parts of the genome, in comparison to single markers or mtDNA sequences, I suggest the authors add a sentence into the discussion about that. This may in the future confirm hypotheses on hybridization gained from the limited data, although of course this also depends on funding.

Some typos:

L191: fecals > fecal

L212: sentence broken

L317/319: who *were* identified

L349: futher > further

L427: not > to

L435: result*ed*

Reviewer #4: This manuscript is about investigating gibbons genetically in the hybrid zones. They make new genetic markers for identifying whether the individual is a “pure-blood” or a “hybrid”. They also discuss on the phenotypic traits for the individuals which were detected as hybrids.

This is a revised manuscript and I was not one of the original reviewers. I will make comments based on the previous reviewers’ comments and the author’s responses.

Overall, the previous comments made by the original reviewers make sense to me and the authors responses are satisfactory. I agree with the previous comments on the limitation of genetic markers. The authors efforts to tone down the conclusion was necessary to make the manuscript more technically sounding.

I have one concern on detecting hybrid using the result of STRUCTURE (e.g. fig 3).

The authors set the criteria for being hybrid as showing contributions of the minor ancestry at a level of more than 0.017. I think it is OK to label the individuals showing “intermediate” level of mixed ancestry as recent hybrids. However, it might be inappropriate to label individuals with marginal value of ancestry (i.e. 0.018 or 0.019) as recent hybrid. This is because the number of markers used is quite limited and even truly “pure-blood” individual may show more than 0.017 of minor ancestry level under this condition.

Then I suggest the author to label the individuals with marginal value of ancestry as “possible hybrids” or so.

7. PLOS authors have the option to publish the peer review history of their article (what does this mean?). If published, this will include your full peer review and any attached files.

Reviewer #3: No

Reviewer #4: No

---

## [Author Response · Author response to Decision Letter 1]

10 Feb 2022

Dear Dr. Gojobori,

Thank you very much for evaluating the revised manuscript entitled “Introgression and mating patterns between white-handed gibbons (Hylobates lar) and pileated gibbons (Hylobates pileatus) in a natural hybrid zone.” We revised our manuscript following comments of reviewers #3 and #4. A major revision point has been that we now adopt the term “possible hybrids” to describe some more ambiguous results in our analyses. We now also mention the limitation of the use of the small number of SNV loci in our study. We hope this revision now resolves reviewers’ concern of potential overstatement of results and thus we toned down some of our findings. Please find our detailed revisions in the reply to reviewers’ comments below.

Reviewer #3: I agree with both previous reviewers that the limited amount of data presented here should not be over-interpreted. In particular, I agree with the statement 1 of reviewer #2, the previously described SNPs might not represent fixed differences with the small number of individuals, especially since ancient admixture events were likely part of their history.

First, the observation in the STRUCTURE analysis could be explained by genetic clines or simply segregating variation rather than recent hybridization. There is one thing I would recommend in terms of analysis, which is a STRUCTURE analysis with higher k (3, 4, 5) to see if the supposed signature of mixing is not just a component restricted to individuals from this area.

Answer:

We performed STRUCTURE analysis with K = 3, 4, and 5. The analyses result almost equally divided “white-handed-gibbon ancestry” into 2, 3, and 4 components in each individual from the three study sites with no detectable change in “pileated-gibbon ancestry.” We interpret this to indicate that the supposed signature of mixing is not just a component restricted to individuals from Khao Yai National Park. We added the new information and analyses with K = 3, 4, and 5 in Methods and Results sections (Lines 248-253 and lines 273-278, respectively).

Second, the threshold for assigning “hybrid ancestry” seems somewhat low and arbitrary given the small number of loci. Inspecting Fig. 3, there are 5-7 likely hybrids with substantial ancestry from both species (if one assumes these loci are really fixed), while I would be more doubtful about the others. The results on morphology and pelage are lending good support to the hypothesis on observed hybrids, especially in the case of those with clear mixed ancestries. I would like to know if the “five hybrid individuals indistinguishable from typical white-handed gibbons” had a lower assignment of admixed ancestry than the three with atypical patterns (L373).

Answer:

There was no clear relationship between the degree of mixed ancestry and the presence/absence of an atypical pelage color pattern among individuals with low levels of mixed ancestry. We added the explanation in Results (Line 379-383).

Third, the discordance of mtDNA nuclear DNA might be pointed out. There are a few individuals with a haplotype assigned to H. pileatus which, in this restricted dataset, have nuclear ancestry only assigned to H. lar. This shows the difficulty in determining ancestries with such few markers. Compare that to humans and Neanderthals, where ancient Neanderthal admixture in modern humans has not been determined by obtaining several mtDNA sequences, nor by retrieving a small amount of nuclear DNA (Green et al., Nature 2006), but only by shallow genome sequencing (Green et al., Science 2010). I agree that it is difficult to obtain information from wild-living primates for this kind of studies, but it is a problem that the confidence in observations must be low when using such a small number of individuals for the reference populations and a very small number of genetic markers. In that sense, I would also add “putative” to any statement about admixture, from the abstract on throughout the paper. I.e. the phrasing should be even more cautious when referring to the genetic results.

Answer:

We followed this and the comment of reviewer #4, and revised corresponding sections to read “putative” when our results seemed ambiguous. Because there was no ambiguity in mixed ancestry among intermediate hybrids, we did not use “putative” to describe ancestry of those individuals. For other gibbons with low levels of mixed ancestry, we decided to recategorized them into “hybrids with low levels of mixed ancestry” and “possible hybrids”. We cannot fully exclude that some arbitrariness for making the decision between those two categories may remain, but considering the morphology and vocalization, the hybrid origin of “hybrids with low levels of mixed ancestry” seems apparent to us thus justifying the use of this category. We believe our revisions resolved the overstatement on the admixture of gibbons in this population.

Finally, technological innovations may likely offer opportunities to obtain much more information from fecal samples of wild primates, as has been shown in chimpanzees with sequencing after capture (see Fontsere et al., Molecular Ecology 2021). Considering the very finegrained picture that were obtained in humans and other primate species after sequencing at least substantial parts of the genome, in comparison to single markers or mtDNA sequences, I suggest the authors add a sentence into the discussion about that. This may in the future confirm hypotheses on hybridization gained from the limited data, although of course this also depends on funding.

Answer:

Following the suggestions, we added the limitation of the use of a small number of SNV markers and the prospects of applying new technologies to fecal genomic analysis of the Khao Yai gibbon population (Line: 443-448); we also added suggested references #55 and #56.

Some typos:

L191: fecals > fecal

L212: sentence broken

L317/319: who *were* identified

L349: futher > further

L427: not > to

L435: result*ed*

Answer:

We corrected all typos.

Reviewer #4: This manuscript is about investigating gibbons genetically in the hybrid zones. They make new genetic markers for identifying whether the individual is a “pure-blood” or a “hybrid”. They also discuss on the phenotypic traits for the individuals which were detected as hybrids.

This is a revised manuscript and I was not one of the original reviewers. I will make comments based on the previous reviewers’ comments and the author’s responses.

Overall, the previous comments made by the original reviewers make sense to me and the authors responses are satisfactory. I agree with the previous comments on the limitation of genetic markers. The authors efforts to tone down the conclusion was necessary to make the manuscript more technically sounding.

I have one concern on detecting hybrid using the result of STRUCTURE (e.g. fig 3).

The authors set the criteria for being hybrid as showing contributions of the minor ancestry at a level of more than 0.017. I think it is OK to label the individuals showing “intermediate” level of mixed ancestry as recent hybrids. However, it might be inappropriate to label individuals with marginal value of ancestry (i.e. 0.018 or 0.019) as recent hybrid. This is because the number of markers used is quite limited and even truly “pure-blood” individual may show more than 0.017 of minor ancestry level under this condition. Then I suggest the author to label the individuals with marginal value of ancestry as “possible hybrids” or so.

Answer:

Please, see our comment in response to reviewer #3 above. In sum, we adopted reviewers’ criticism and now introduced “possible hybrids” for individuals with marginal mixed ancestry.

---

## [Decision Letter · Decision Letter 2]

14 Feb 2022

Introgression and mating patterns between white-handed gibbons (Hylobates lar) and pileated gibbons (Hylobates pileatus) in a natural hybrid zone

PONE-D-21-19052R2

Dear Dr. Matsudaira,

We’re pleased to inform you that your manuscript has been judged scientifically suitable for publication and will be formally accepted for publication once it meets all outstanding technical requirements.

Kind regards,

Jun Gojobori

Academic Editor

PLOS ONE

Additional Editor Comments (optional):

Reviewers' comments:

Reviewer's Responses to Questions

**Comments to the Author**

1. If the authors have adequately addressed your comments raised in a previous round of review and you feel that this manuscript is now acceptable for publication, you may indicate that here to bypass the “Comments to the Author” section, enter your conflict of interest statement in the “Confidential to Editor” section, and submit your "Accept" recommendation.

Reviewer #3: All comments have been addressed

Reviewer #4: All comments have been addressed

2. Is the manuscript technically sound, and do the data support the conclusions?

Reviewer #3: Yes

Reviewer #4: (No Response)

3. Has the statistical analysis been performed appropriately and rigorously? 

Reviewer #3: Yes

Reviewer #4: (No Response)

4. Have the authors made all data underlying the findings in their manuscript fully available?

Reviewer #3: Yes

Reviewer #4: (No Response)

5. Is the manuscript presented in an intelligible fashion and written in standard English?

Reviewer #3: Yes

Reviewer #4: (No Response)

6. Review Comments to the Author

Reviewer #3: (No Response)

Reviewer #4: (No Response)

7. PLOS authors have the option to publish the peer review history of their article (what does this mean?). If published, this will include your full peer review and any attached files.

Reviewer #3: No

Reviewer #4: No

---

## [Editor Report · Acceptance letter]

23 Mar 2022

PONE-D-21-19052R2 

Introgression and mating patterns between white-handed gibbons (*Hylobates lar*) and pileated gibbons (*Hylobates pileatus*) in a natural hybrid zone 

Dear Dr. Matsudaira:

I'm pleased to inform you that your manuscript has been deemed suitable for publication in PLOS ONE. Congratulations! Your manuscript is now with our production department. 

Kind regards, 

on behalf of

Dr. Jun Gojobori 

Academic Editor

PLOS ONE